# ManiCast: Collaborative Manipulation with Cost-Aware Human Forecasting

**Kushal Kedia**
Cornell University

**Prithwish Dan**
Cornell University

**Atiksh Bhardwaj**
Cornell University

**Sanjiban Choudhury**
Cornell University
*

**Abstract:** Seamless human-robot manipulation in close proximity relies on accurate forecasts of human motion. While there has been significant progress in learning forecast models at scale, when applied to manipulation tasks, these models accrue high errors at critical transition points leading to degradation in downstream planning performance. Our key insight is that instead of predicting the most likely human motion, it is sufficient to produce forecasts that capture how future human motion would affect the cost of a robot's plan. We present MANI-CAST, a novel framework that learns cost-aware human forecasts and feeds them to a model predictive control planner to execute collaborative manipulation tasks. Our framework enables fluid, real-time interactions between a human and a 7-DoF robot arm across a number of real-world tasks such as reactive stirring, object handovers, and collaborative table setting. We evaluate both the motion forecasts and the end-to-end forecaster-planner system against a range of learned and heuristic baselines while additionally contributing new datasets. We release our code and datasets at https://portal-cornell.github.io/manicast/.

**Keywords:** Collaborative Manipulation, Forecasting, Model Predictive Control

## 1 Introduction

Seamless collaboration between humans and robots requires the ability to accurately anticipate human actions. Consider a shared manipulation task where a human and a robot collaborate to cook a soup – as the robot stirs the pot, the human adds in vegetables. Such close proximity interactions require fluid adaptions to the human partner while staying safe. To do this, the robot must predict or forecast the human's arm movements, and plan with such forecasts. This paper addresses the problem of generating human motion forecasts that enable seamless collaborative manipulation.

Recent works have made considerable progress in training human motion forecast models by leveraging large-scale human activity datasets such as AMASS [1] and Human 3.6M [2]. However, directly applying such models for human-robot collaboration presents several challenges. First, the space of all possible human motions is very large and pre-trained forecast models typically average out their performance over the distribution of activities seen at train time. Second, although fine-tuning with task-specific data helps improve overall forecasting accuracy, it does not necessarily lead to better performance for downstream planning. This because these models are usually very accurate at predicting frequent and predictable events in the data, e.g. continuing to stir a ladle. However, they struggle with rare and more unpredictable transitions, e.g. pulling the ladle back as a human hand enters the workspace. Such transitions are critical for seamless collaboration and errors at such data points have a substantial impact on the overall performance of the robot.

***Our key insight is that instead of predicting the most likely human motion, it is sufficient to produce forecasts that capture how future human motion would affect the cost of a robot's plan.*** For instance, in the cooking task, the robot's planned trajectory has high cost if it comes close to the human arm, and low cost otherwise. While trying to accurately predict how the human arm may move at any given moment is difficult, it is much easier to predict whether that movement results in a high cost. We achieve this by modifying the forecast training objective to match the cost of

---

*{kk837, pd337, ab2635, sc2582}@cornell.edu

7th Conference on Robot Learning (CoRL 2023), Atlanta, USA.

| **Reactive Stirring** | **Object Handover** | **Collaborative Table Setting** |
|---|---|---|

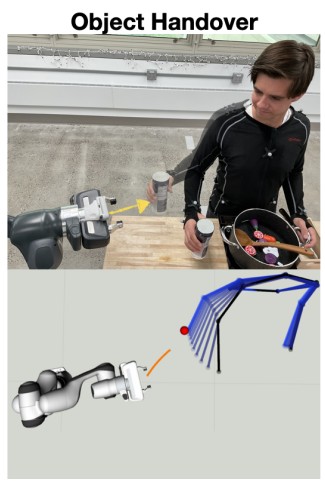
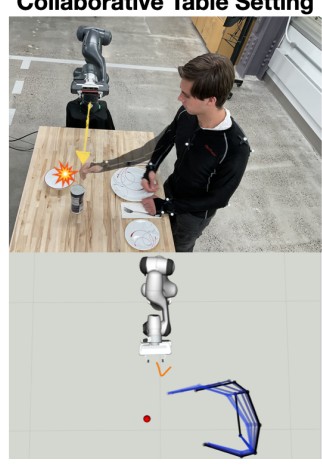

Figure 1: Closed-loop, real-time collaborative human-robot manipulation across three different kitchen tasks by combining learned human pose forecasts with model predictive control.

ground truth future motions rather than the exact motion itself. We propose a novel framework, MANICAST (Manipulation Forecast), that learns cost-aware human motion forecasts and plans with such forecasts for collaborative manipulation tasks. At train time, we fine-tune pre-trained human motion forecasting models on task specific datasets by upsampling transition points and upweighting joint dimensions that dominate the cost of the robot's planned trajectory. At inference time, we feed these forecasts into a model predictive control (MPC) planner to compute robot plans that are reactive and keep a safe distance from the human. To the best of our knowledge, this is the first paper to leverage large-scale human motion data and state-of-the-art pre-trained human forecast models to integrate with a real-time MPC planner for collaborative human-robot manipulation tasks. Our key contributions are:

1. A method to train human motion forecast models to be used with a downstream planner for collaborative human-robot manipulation.
2. A new dataset of human-human collaborative manipulation on 3 kitchen tasks.
3. Real-world evaluation of a human-robot team on each of the 3 tasks and comparison against a range of learned and heuristic baselines.

## 2 Related Work

**Navigation with Human Forecasts.** Navigating around humans has been a long-standing challenge in areas such as self-driving [3–8] and social robotics [9–15]. In the context of self-driving, there's a rich history in trajectory forecasting of agents observed by the autonomous vehicle (AV) [16] ranging from physics-based methods (e.g. Constant Velocity/Acceleration [17–21], Kalman Filter [22–24]) to machine learning-based methods (e.g. Gaussian Processes [25, 26], Hidden Markov Models [27, 28]) to inverse reinforcement learning [29, 30]. In recent years, the rise of sequence models in deep learning has led to great advances in accurately predicting future states of traffic participants. Parallel to the developments in autonomous driving, social navigation in crowd environments has similarly converged on sequence models [31–35] to forecast trajectories within a crowd. In addition to making predictions about trajectories, many recent works have used forecasts as inputs to motion planners [36–38] or jointly forecasted and planned [39, 40]. The locations swept out by forecasts are considered to be "unsafe" collision regions that the planner avoids. Consequently, evaluation of trajectory forecasting has shifted from accuracy-based metrics [41] to more task-aware metrics that focus on the performance of a downstream planner. However, all of these works approach forecasting as a 2D problem, whereas we look to plan 7D robot manipulation trajectories by forecasting 21D human pose, which is higher dimensional and much more complex.

**Human Pose Forecasting.** Human pose forecasting involves predicting how a human pose evolves in both space and time. Several works have leveraged Recurrent Neural Networks (RNN) [42, 43] and Transformer Networks [44] to model time, while graph-based models such as Graph Convolutional Networks (GCN) [45, 46] have been used to model the interaction of joints in space. Mao et al. [45] note that human motion over a time horizon is smooth and periodic. They exploit this observation by encoding the context of an agent with a Discrete Cosine Transform (DCT) and then apply

an attention module to capture the similarities between current and historical motion sub-sequences. STS-GCN [46] learns the adjacency matrix between different joints and across different timesteps separately to bottleneck the cross-talk interactions between joints across time and use a Temporal Convolutional Network (TCN) for decoding the predictions over a fixed time horizon. Recent works have extended forecasting from single-person to multi-person forecasting [47, 48]. Although there have been many advances in pose forecasting, to the best of our knowledge, our work is the first to integrate an entire upper body forecast into a robot manipulation planner.

**Close-Proximity Human Robot Collaboration** For close proximity human-robot tasks where fluidity is important, forecasting human pose is essential. While there are various works that forecast human pose in the context of robotics [49–53], they don't integrate such predictions into robot planners. Research focused on robot manipulation planning [54–56] around humans has modeled humans in some form at planning time, whether that be assuming static poses [56, 57], tracking the current pose [58, 59], or making predictions only about the motion of specific joints such as the wrist or head [60–63]. Mainprice et al. [64] predicts single-arm reaching motion using motion capture of two humans performing a collaborative task in a shared workspace and learns a collision avoidance cost function using Inverse Optimal Control (IOC). He et al. [65] introduce a hierarchical approach with a high and low-level controller to generate feasible trajectories and ensure safety. Scheele et al. [60] use an unsupervised online learning algorithm to build a model that predicts the remainder of a human reaching motion given the start of it. Ling et al. [61] use the human's head pose, wrist position, and wrist speed as inputs to their forecasting Long Short-Term Memory (LSTM) model to predict future wrist positions which are used as an input to their robot motion planner. Oguz et al. [66] propose a framework to detect and classify interactions during close-proximity human-robot interactions. Prasad et al. [67] propose a framework to plan humanoid motion plans representing human motion as a latent variable. In contrast to these methods, we utilized orecasting models pre-trained on larger, more diverse data to solve human-robot collaboration tasks. We focus on the efficacy of the forecaster, especially in extended interaction settings where performance in transition windows is critical but forms a small part of our collected dataset.

## 3   Problem Formulation

**Notation.** The human's state at timestep $t$, $s_t^H \in \mathbb{R}^{J \times 3}$, are the 3-$D$ coordinates of $J$ upper-body joints. Let the context[2], $\phi = \{s_{-k+1}^H, \ldots, s_0^H\}$, be the history of human states over the past $k$ timesteps. Let future human and robot trajectories over a horizon $T$ be $\xi_H = \{s_1^H, s_2^H, \ldots, s_T^H\}$ and $\xi_R = \{s_1^R, s_2^R, \ldots, s_T^R\}$ respectively. Let $C(\xi_R|\xi_H)$ denote the cost of the robot trajectory given the human trajectory. Let $P_\theta(\xi_H|\phi) \propto \exp(-\|\xi_H - \mu_\theta(\phi)\|)$ be the learnt forecast model, modelled as a Gaussian, where $\mu_\theta(\phi)$ is the predicted mean trajectory, and $\theta$ are the learned parameters.

**Planning.** The goal of the planner is to compute the optimal robot trajectory $\xi_R$ given human trajectory $\xi_H$. At inference time, $\xi_H$ is unknown, and the planner must rely on a *forecast* $\hat{\xi}_H \sim P_\theta(.|\phi)$ of the human trajectory generated by the model. The planner then solves for the lowest cost robot trajectory given the forecast — $\arg\min_{\xi_R} \mathbb{E}_{\hat{\xi}_H \sim P_\theta(.|\phi)} C(\xi_R|\hat{\xi}_H)$.

**Cost-Aware Human-Pose Forecasting.** Typically, the forecast model is trained by maximizing log-likelihood (MLE) of the ground truth human trajectory $\xi_H$, i.e., $\max_\theta \log P_\theta(\xi_H|\phi)$, which for a Gaussian model corresponds to a L2 loss. However, low log-loss can still result in a high mismatch between the costs $C(.|\xi_H)$ and $C(.|\hat{\xi}_H)$, which can result in poor downstream planner performance. For instance, in Fig. 4, the "base model" minimizing log-loss has low errors everywhere except at critical transition points, which has a significant impact on the cost function. Instead, we choose a loss function $\ell(\theta)$ such that the costs calculated with the forecasts match the cost calculated with ground truth for any robot plan $\xi_R$:

$$\ell(\theta) = \mathbb{E}_\phi \left[ \left| \mathbb{E}_{\xi_H \sim P(.|\phi)} C(\xi_R|\xi_H) - \mathbb{E}_{\hat{\xi}_H \sim P_\theta(.|\phi)} C(\xi_R|\hat{\xi}_H) \right| \right] \tag{1}$$

This loss is equivalent to matching moments of a cost function [68, 69]. However, directly implementing this as a loss function is challenging in practice for a number of reasons. First, the cost

---

[2]The context can have other factors like the history of the robot's past states, which we exclude for simplicity

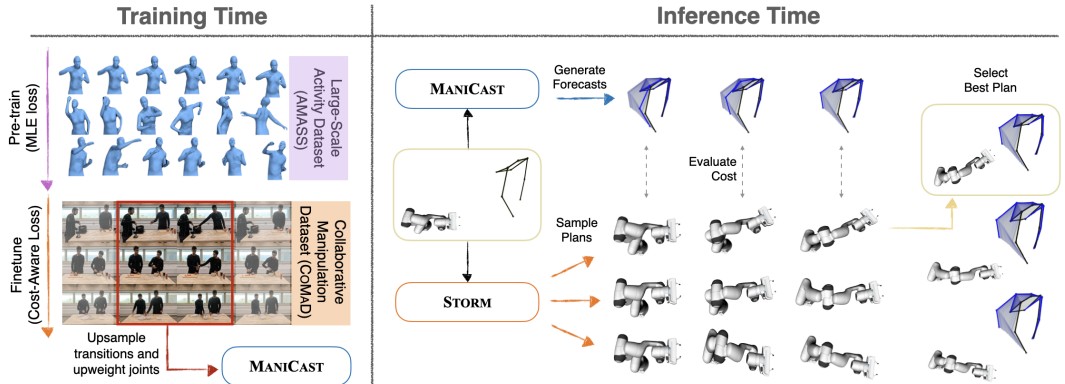

Figure 2: Overview of training and inference pipeline for MANICAST. During training, we pre-train MAN-ICAST on a large human activity database and finetune on CoMaD using a cost-aware loss by upsampling transitions and upweighting wrist joints. During inference, we use a sampling-based MPC, STORM, to rollout multiple robot trajectories, evaluate their costs with MANICAST forecasts and select the best plan.

function has non-differentiable components like collision check modules. Second, the cost function may not be convex and have poor convergence rates. Finally, the cost function may not be known upfront at train time, and likely to be tuned frequently. We address these challenges next.

## 4 Approach

We present MANICAST, a framework that learns cost-aware human forecasts and plans with such forecasts for collaborative manipulation tasks. Fig. 2 shows both the training time and inference time process. At train time, we pre-train a state-of-the-art forecast model (STS-GCN [46]) on large-scale human activity data (AMASS [1]) and fine-tune the forecasts on our Collaborative Manipulation Dataset (CoMaD), with a focus on optimizing downstream planning performance. At inference time, we detect the human pose at 120Hz using an Optitrack motion capture system, generate forecasts at 120Hz, feed the forecasts into a MPC planner (STORM [70]) to compute robot plans at 50 Hz.

### 4.1 Train Time: Fine-tune Forecasts with Cost-Aware Loss

We first pre-train the model using standard MLE loss on a large scale human activity dataset to get a baseline forecast model. We then fine-tune this model on our own Collaborative Manipulation Dataset (CoMaD), that has data of two humans collaboratively performing manipulation tasks. Since directly optimizing the cost-aware loss in (1) is challenging, we propose two strategies to approximately optimize the loss without significant changes to the model training pipeline.

**Strategy 1: Importance Sampling.** While the MLE loss measures errors uniformly on the distribution $P(\phi)$, the cost-aware loss $\ell(\theta)$ (1) is sensitive to errors at contexts where costs are generally higher. Notably, for many tasks, transition points where the human comes into the robot's workspace are typically high cost and hence dominate the loss. Since transition points are infrequent, the fine-tuned model has higher errors on them. Our strategy is to assign greater importance to transition points by importance sampling and then finetuning on a new distribution, which we formalize below.

Let $C_{\max}(\phi)$ be the maximum cost of a robot trajectory that a given context can induce, which we compute from collected data. We defined a *transition distribution* as $P_T(\phi) \propto P(\phi)\mathbb{I}(C_{\max}(\phi) \geq \delta)$, i.e., the distribution over contexts that induce a cost higher than a percentile threshold $\delta$. Instead of minimizing the loss on $P(\phi)$, we choose a new distribution $Q(\phi) = 0.5P(\phi) + 0.5P_T(\phi)$ that mixes the original distribution with the transition distribution, effectively up-sampling the transitions. We prove the following performance bounds.

**Lemma 1.** *For a model $\theta$ with bounded loss of $\varepsilon$ on $P(\phi)$, the final loss is bounded as $\ell(\theta) \leq C_{\max}\varepsilon$, where $C_{\max} = \max_\phi C_{\max}(\phi)$. In contrast, for a model with bounded loss of $\varepsilon$ on the new distribution $Q(\phi)$, the final loss is bounded by $\ell(\theta) \leq 2\max(\delta, C_{\max}\mathbb{E}_{P(\phi)}[\mathbb{I}(C_{\max}(\phi) \geq \delta)])\varepsilon$*

We refer the reader to the Appendix for the proof and interpretation of the bound. For our tasks, we choose a small $\delta$ (10%) that works well in practice.

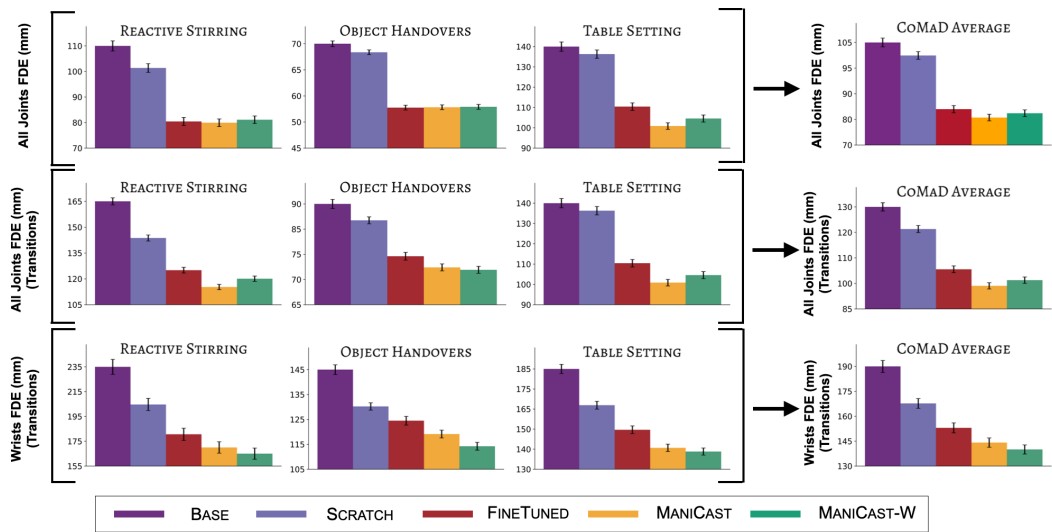

Figure 3: Forecasting Metrics (All Joints FDE, T-All Joints FDE, T-Wrist FDE) across all tasks in CoMaD.

**Strategy 2: Dimension Weighting.** While the MLE or L2 loss sets equal weights to all joint dimensions $J$, the cost-aware loss $\ell(\theta)$ (1) is sensitive to errors along certain dimensions. For example for a handover task, a small error in predicting the wrist position can have a large impact on the cost. We empirically tune weights $w \in \mathbb{R}^J$ on the MLE loss to upweight joints that are explicitly used as terms in the cost functions we define.

We combine these two strategies to optimize the following proxy loss:

$$\hat{\ell}(\boldsymbol{\theta}) = \mathbb{E}_{0.5P(\phi)+0.5P_T(\phi)} \left[ \mathbb{E}_{\xi_H \sim P(.|\phi)} \sum_j^J w_j \log P_\theta(\xi_H^j | \phi) \right] \tag{2}$$

### 4.2 Inference Time: Sampling-Based Model Predictive Control with Learned Forecasts

At inference time, we invoke a sampling based model predictive control (STORM [70]) to compute plans for a 7-DoF robot arm. We designed a set of three cost functions, one for each collaborative manipulation task, that take as input a candidate robot plan and the forecasts generated by MAN-ICAST and computes a cost. At every timestep, given a context $\phi$, the model predictive control samples plans, evaluates the cost of each plan and updates the sampling distribution till convergence, returning the minimum cost plan. The robot takes a step along the plan and replans. We provide details on the cost function and the MPC planner in the Appendix.

## 5 Experiments

### 5.1 Experimental Setup

**Collaborative Manipulation Dataset (CoMaD).** We design 3 collaborative manipulation tasks shown in Fig 1 (1) *Reactive Stirring*: Robot stirs a pot while making way for human pouring in vegetables (2) *Object Handovers*: Robot moves to receive an object behind handed over by the human (3) *Collaborative Table Setting*: robot and the human manipulate objects on a table while not getting in each other's way. To train our forecasting model, we collect a dataset of two humans executing these tasks. It contains 19 episodes of reactive stirring, 27 episodes of handovers, and 15 episodes of collaborative table setting. We refer to Appendix for more details.

**Large Human-Activity Databases.** The AMASS (Archive of Motion Capture As Surface Shapes) [1] dataset is a large and diverse collection of human motion. It consists of over 40 hours of single-human motion spanning over 300 subjects. Hence we pre-train models on AMASS.

**Forecast Models.** We run the planner with different forecasts. (1) *Ground Truth*: CUR assumes the current human pose remains constant over the planning horizon, and FUT uses the real fu-

| Model → | GROUNDTRUTH | BASELINES | LEARNING-BASED | | | |
|---|---|---|---|---|---|---|
| Metric ↓ | CUR | CVM | FINETUNED | MANICAST-T | MANICAST | MANICAST-W |
| **REACT** Stop Time (ms) | 0 ($\pm$0) | 367.8 ($\pm$50.9) | 203.3 ($\pm$22.3) | 271.1 ($\pm$25.6) | 246.7 ($\pm$25.8) | 290.0 ($\pm$29.9) |
| Restart Time (ms) | 0 ($\pm$0) | 235.6 ($\pm$18.2) | 441.1 ($\pm$30.8) | 454.4 ($\pm$36.7) | 455.6 ($\pm$33.5) | 496.7 ($\pm$27.9) |
| FDR | 0% | 67% | 0% | 11% | 0% | 11% |
| **HAND.** Goal Detection (ms) | 0 ($\pm$0) | 246.0 ($\pm$146.0) | 189.3 ($\pm$47.8) | 473.3 ($\pm$123.7) | 450.0 ($\pm$124.8) | 488.4 ($\pm$133.4) |
| Correct Goal Rate | 100% | 20% | 100% | 100% | 100% | 100% |
| Path Length (mm) | 459.0 ($\pm$37.0) | 485.0 ($\pm$85.0) | 432.0 ($\pm$39.0) | 428 ($\pm$42.0) | 404.0 ($\pm$45.0) | 436.0 ($\pm$49.4) |
| Time to Goal (s) | 4.59 ($\pm$0.37) | 3.77 ($\pm$0.86) | 3.87 ($\pm$0.28) | 3.91 ($\pm$0.32) | 3.96($\pm$0.54) | 4.17 ($\pm$0.45) |

Table 1: We integrate forecasts of different models into STORM for the reactive stirring and object handover tasks. The standard error for each planning metric is shown inside parentheses.

tures obtained from playing back CoMaD's episodes. While the latter can not be used for real-world closed-loop planning, it's the gold standard for comparing other models. (2) *Baselines*: The Constant Velocity Model (CVM) calculates future human pose by integrating a constant velocity, calculated from the immediate 0.4s history of each joint. The WORST case model is a conservative model that constructs two large spheres (arm length radius) centered at each shoulder joint. (3) LEARNING-BASED: We train our forecasting models using the STS-GCN [46] architecture. The BASE model is trained only on AMASS data, whereas the SCRATCH model is trained only on CoMaD. The FINE-TUNED model pre-trains on AMASS, but is finetuned on CoMaD only using the MLE objective. MANICAST is finetuned using the loss objective in Eq.2 with the weights $w_j = 1$. MANICAST-T is finetuned only on transition data in CoMaD. MANICAST-W upweights loss for wrist joints by 5x.

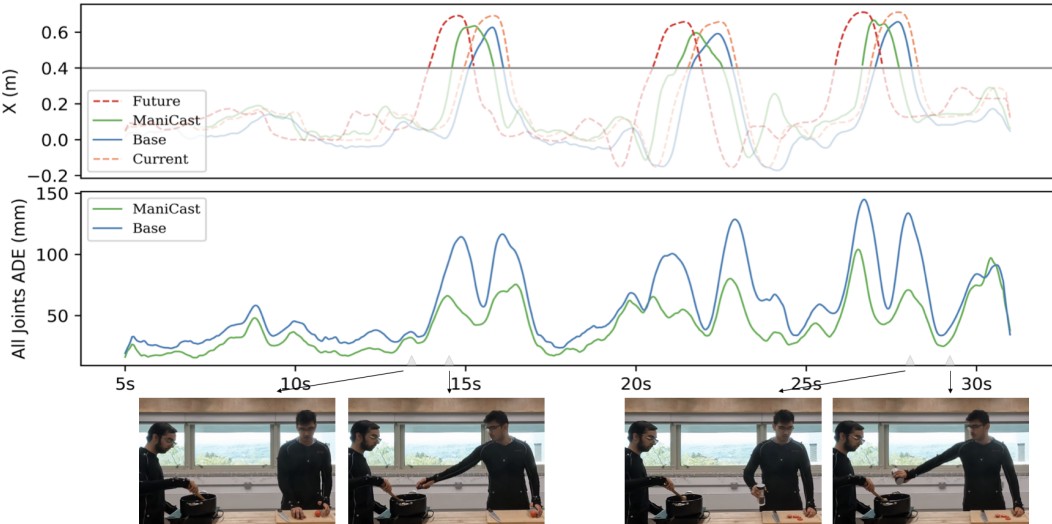

Figure 4: (Top) The *x*-position of the reaching human's wrist in a Reactive Stirring test set episode. $x \geq 0.4$ indicates the wrist is near the pot. (Bottom) BASE model's forecasts have high errors during transitions and lag behind the current pose. MANICAST, trained on CoMaD by upsampling transitions, predicts the reaching human's pose faster than tracking the current pose.

**Forecasting Metrics.** We measure the Average Displacement Error (ADE) across all 25 timesteps of prediction and the Final Displacement Error (FDE) at the final timestep of prediction. Metrics are measured separately for 7 upper body joints and wrist joints. We also report the wrist errors inside the transition windows for the stirring and handover tasks.

**Planning Metrics.** For quantitative results, we ran a 7-DoF Franka Emika Research 3 in the real-world and played back a recording of a human partner from CoMaD. *Reactive Stirring*: Test set had 9 human movements into the pot. We report average time required by the robot to stop and restart motion compared to using the current position for planning. We also report the False Detection Rate (FDR) for the human coming into the robot's workspace. *Object Handover*: Test set had 10 handovers. We report the average gain in Goal Detection time compared to using the current pose and the percentage of times when the correct goal is detected by the forecaster. Among instances where the correct goal is detected, we report the average time the robot arm took to reach the handover location and the total path distance moved by its end-effector. For qualitative closed-loop evaluation (Fig 1, Fig 5, Fig 6), we run our entire pipeline with two new human subjects **not part of CoMaD**.

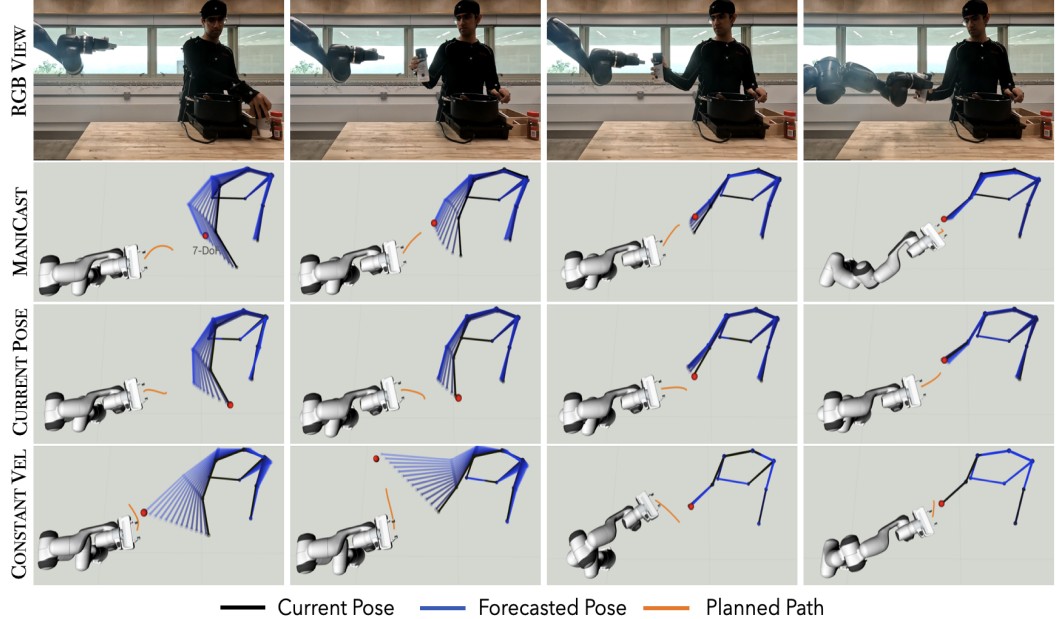

Current Pose — Forecasted Pose — Planned Path

Figure 5: Integrating forecasting models with MPC for real-world human-robot handovers. MAN-ICAST forecasts allow the robot to track a shorter path to the handover location in less time than following the current pose or the unrealistic constant velocity model's predictions.

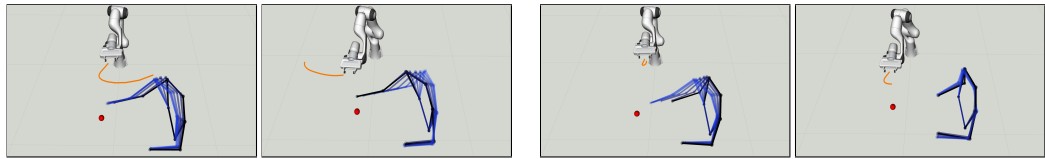

Planning with current pose tracking      Planning with MANICAST forecasts

Figure 6: Collision avoidance in Collaborative Table Setting. Following the current pose, the robot arm nearly collides with the human due to delayed prediction. MANICAST avoids this by slowing down when the human reaches in and speeding up when the human retracts their arm.

## 5.2 Results and Analysis

*O1.* **MANICAST forecasts outperform current pose tracking.** Figure 2 shows that MANICAST models have low FDE in forecasting human pose with improved planning performance (Table 1). In the reactive stirring task, MANICAST models predict both the arrival and departure of humans before the current pose. Fig 4 showcases MANICAST predictions over an entire episode from the CoMaD's test set. In the object handover task, current pose tracking predicts the goal location slower than the MANICAST models. As Fig 5 demonstrates, the robot's end effector chases the current wrist pose towards its eventual final location, leading to a longer trajectory that takes more time to execute. Further, planning with forecasts leads to safe motion as collisions can be prevented with future human positions (Fig 6).

*O2.* **BASELINE models predict dynamically infeasible forecasts leading to suboptimal planning performance.** As seen in Fig 5, since the CVM's predictions are not dynamically constrained, its forecasts overpredict future human positions making the robot arm deviate from the optimal path during handovers. On the other hand, as seen in Table 1 in the Appendix, the robot arm remains retracted during the reactive stirring task and does not approach the human's wrist for handovers following the extremely conservative WORST case model.

*O3.* **Training on AMASS or CoMaD alone is not sufficient to achieve optimal performance.** The BASE and SCRATCH models produce erroneous forecasts on CoMaD (Fig 3). In most cases, they have larger prediction errors than simply tracking the current pose. While the BASE model produces the most accurate forecasts on AMASS (Tab 2 in Appendix), the activities in CoMaD contain reaching arm motions and abrupt transitions that are not captured by the model. Fig 4 shows that its predictions lag behind current pose, when predicting the arm to retract. Whereas, SCRATCH

is only trained on CoMaD. While this model is exposed to task-specific data, its training does not converge as forecasting 21D human pose is a data-intensive complex task. Consequently, planning with both of these forecasters leads to worse performance than just tracking the current pose (Table 1).

*O4.* **By upsampling transition points, MANICAST improves the accuracy and efficiency of task completion.** Observing Fig 3, we note that MANICAST models have higher displacement errors on CoMaD's test set than the FINETUNED model trained on just the MLE objective. However, since the MANICAST models upsample transitions, they have lower prediction errors in transition windows relevant to planning performance. In the reactive stirring task, they detect the arrival and departure of the human in the robot's workspace quicker than FINETUNED. In the object handover task, this difference is pronounced as the MANICAST models predict the final handover location more than 2 times faster than FINETUNED (Table 1). MANICAST-T is trained on only transitions and planning with it can lead to erratic performance in some cases. For example, in the reactive stirring task, it falsely predicts the human's arrival into the robot's workspace in a test set episode.

*O5.* **Upweighting wrist joints in the loss function can improve performance in planning tasks.** For the collaborative manipulation tasks considered in this work, we note that predicting the location of the wrist joints has higher priority than the other upper body joints. MANICAST-W assigns 5 times more weight to the wrist joints in its loss function. This leads to lower forecasting errors for the wrist joint in CoMaD compared to all other models (Fig 3). In most cases (Table 1), this model has the best planning performance with faster detection of human pose in the robot's workspace during both the reactive stirring and handover tasks. In some cases, we note that its predictions can be unstable leading to one false detection in the reactive stirring task and larger end-effector path movement in the handover task.

## 6 Discussion

This work considers the problem of collaborative robot manipulation in the presence of humans in the workspace. We present MANICAST, a novel framework for seamless human-robot collaboration that generates cost-aware forecasts and plans with them in real-time. Our approach was thoroughly tested on three collaborative tasks with a human partner in a real-world setting. MANICAST **leverages large databases of human activity**. Producing dynamically consistent 21D human pose is a complex and high-dimensional problem. However, by training on a large dataset, ManiCast is able to learn the statistical regularities of human motion. MANICAST **optimizes planning performance** by using an approximate cost-aware loss function. We finetune the forecasting model on a novel Collaborative Manipulation Dataset (CoMaD) by upsampling transition points and upweighting relevant joint dimensions. Our system can track a human user's pose, forecast their movements, and control a robotic arm in **real-time and at high speed**. Our system's forecasting module runs at 120Hz, used by an MPC at 50Hz. Fast replanning enables robots to collaborate safely with humans by allowing them to adjust their motion in response to abrupt changes in their environment. In future directions of work, we plan to extend our framework to produce conditional forecasts of human motion given robot trajectories. Further, we will attempt to directly optimize the cost function for a task instead of approximating the forecasting loss function.

## 7 Limitations

Several reasons limit our approach's deployment to real-world human users. Firstly, our method relies on a motion capture system consisting of 10 cameras to track the history of a human's pose. Further, the user is required to wear a motion capture suit which can be cumbersome. Future work will attempt to detect human skeletons using an ego-centric camera view. Secondly, CoMaD captures just two human subjects across its entire dataset. While we qualitatively show generalization to a new human subject, we do not extensively test it on a range of users. Additionally, the forecaster may not generalize to users with movement styles not represented in the data. Future datasets should encompass multiple subjects with diverse movement styles for widespread usage of human-pose forecasting in personal robotics. This must be accompanied by a comprehensive user study to test the overall pipeline. Finally, we only considered the upper body skeleton for forecasting motion as there was significant occlusion in detecting the lower body. Our forecaster can not be directly applied to robotics applications in which the human's lower body movement is of interest.

**Acknowledgments**

This work was partially funded by NSF RI (#2312956).

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

# A Appendix

We run additional experiments, especially focusing on the importance of wrist prediction. We also provide the proof for lemma 1 and include additional details about the MPC planner, the collaborative manipulation tasks, our dataset, and model implementation.

## A.1 Additional Baselines

We report forecasting metrics across AMASS and CoMaD datasets in Table 2 and planning metrics, including more baselines in Table 3. Section 5.1 explains the different model implementations.

## A.2 Focusing on Wrist Predictions

In this section, we experiment with different wrist weights for MANICAST-W models. Further, we compare with a variant of the FINETUNED that upweights wrist dimensions, FINETUNED-W (MLE + Wrist Weighting).

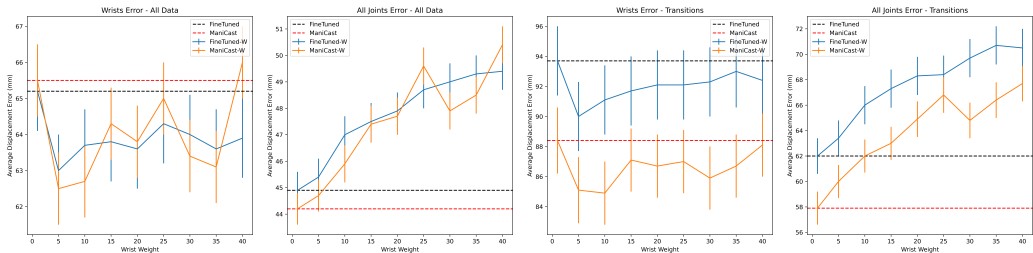

Figure 7: (Reactive Stirring) Forecasting Errors (with error bars) for different wrist weights.

Observing the graphs in Fig 7, we note that all MANICAST-W models that upsample transition data have lower all joints and wrists ADE in the transition periods than the FINETUNED-W model that upweights wrist errors on the MLE loss. We observe a trade-off exists between all joints' and wrists' forecasting errors. An increase in the wrist weight in MANICAST-W reduces prediction errors on the wrist joints, but at the same time, the prediction errors on all joints increase. Furthermore, the decrease in wrist errors eventually plateaus around a wrist weight of 5, which justifies this hyperparameter choice for the MANICAST-W model presented in the main paper.

## A.3 Fitting a Typical Pose to a Wrist Only Forecast

We consider the simpler problem of only predicting the wrist joint position in the Reactive Stirring Task. We still utilize the entire upper body history as input to the forecasting model which we name WRISTONLY. We construct the rest of the upper body by assuming the upper body at the last observable timestep remains still in the future. We consider a variant of the model that upsamples transition points, naming it WRISTONLY-T.

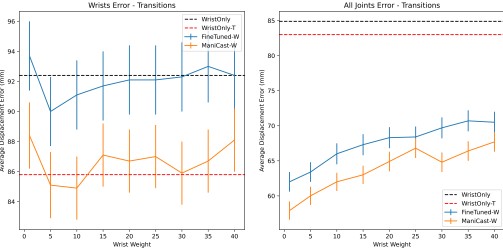

Figure 8: (Reactive Stirring) Comparing Forecasting Errors with TYPICALPOSE model

In Fig 8, we observe that wrist forecasting errors for the WRISTONLY and WRISTONLY-T are similar to the FINETUNED-W and MANICAST-T models that also predict the rest of the human upper body. We will still find that upsampling transition data points helps reduce the forecasting error.

| | Metrics (mm) ↓ | BASE | SCRATCH | FINETUNED | MANICAST-T | MANICAST | MANICAST-W |
|---|---|---|---|---|---|---|---|
| **AMASS** | All Joints ADE | 63.0 (±0.1) | 181.2 (±0.2) | 99.0 (±0.1) | 103.5 (±0.1) | 103.5 (±0.1) | 99.1 (±0.1) |
| | All Joints FDE | 92.1 (±0.2) | 196.7 (±0.2) | 136.8 (±0.2) | 141.6 (±0.2) | 141.5 (±0.2) | 135.3 (±0.2) |
| | Wrists ADE | 103.9 (±0.2) | 263.0 (±0.3) | 157.5 (±0.3) | 160.0 (±0.3) | 157.2 (±0.3) | 156.0 (±0.3) |
| | Wrists FDE | 154.8 (±0.4) | 275.4 (±0.4) | 218.6 (±0.4) | 219.7 (±0.4) | 212.8 (±0.4) | 214.4 (±0.4) |
| **REACTIVESTIR** | All Joints ADE | 67.7 (±1.0) | 67.4 (±0.9) | 44.8 (±0.7) | 50.6 (±0.6) | 45.1 (±0.6) | 45.2 (±0.6) |
| | All Joints FDE | 110.8 (±2.0) | 102.1 (±1.7) | 81.0 (±1.6) | 91.3 (±1.5) | 80.5 (±1.5) | 81.7 (±1.5) |
| | Wrists ADE | 94.8 (±1.4) | 93.6 (±1.3) | 64.7 (±1.1) | 75.9 (±1.0) | 67.1 (±1.0) | 62.9 (±1.0) |
| | Wrists FDE | 154.2 (±2.8) | 137.9 (±2.3) | 113.2 (±2.3) | 131.7 (±2.1) | 115.0 (±2.2) | 110.2 (±2.1) |
| | T-All Joints ADE | 92.2 (±2.1) | 84.7 (±1.7) | 63.3 (±1.5) | 60.7 (±1.3) | 58.4 (±1.3) | 60.7 (±1.3) |
| | T-All Joints FDE | 163.8 (±4.1) | 143.1 (±3.5) | 124.9 (±3.2) | 120.3 (±3.0) | 115.4 (±3.0) | 120.1 (±3.0) |
| | T-Wrists ADE | 134.3 (±3.3) | 129.2 (±2.7) | 94.9 (±2.4) | 91.1 (±2.1) | 89.5 (±2.2) | 85.3 (±2.1) |
| | T-Wrists FDE | 233.8 (±6.0) | 203.5 (±4.9) | 179.5 (±4.9) | 171.3 (±4.4) | 168.9 (±4.6) | 163.9 (±4.5) |
| **HANDOVER** | All Joints ADE | 45.2 (±0.3) | 49.9 (±0.3) | 31.8 (±0.3) | 34.1 (±0.3) | 32.5 (±0.3) | 32.4 (±0.3) |
| | All Joints FDE | 70.4 (±0.7) | 68.3 (±0.6) | 54.6 (±0.6) | 57.5 (±0.6) | 54.7 (±0.6) | 54.8 (±0.6) |
| | Wrists ADE | 62.0 (±0.5) | 68.4 (±0.5) | 46.8 (±0.5) | 50.7 (±0.4) | 48.2 (±0.4) | 45.8 (±0.4) |
| | Wrists FDE | 100.1 (±1.0) | 92.6 (±0.8) | 79.0 (±0.9) | 84.5 (±0.8) | 79.4 (±0.8) | 77.2 (±0.8) |
| | T-All Joints ADE | 55.3 (±0.7) | 61.7 (±0.6) | 42.1 (±0.6) | 41.3 (±0.5) | 41.0 (±0.5) | 40.7 (±0.5) |
| | T-All Joints FDE | 91.9 (±1.3) | 87.2 (±1.0) | 76.9 (±1.1) | 74.8 (±1.1) | 73.7 (±1.0) | 73.0 (±1.0) |
| | T-Wrists ADE | 84.8 (±1.2) | 91.2 (±1.0) | 67.6 (±1.0) | 64.8 (±0.9) | 65.8 (±0.9) | 62.6 (±0.9) |
| | T-Wrists FDE | 146.7 (±2.1) | 130.7 (±1.6) | 124.5 (±1.9) | 118.3 (±1.7) | 118.7 (±1.7) | 113.4 (±1.7) |
| **TABLESET** | All Joints ADE | 80.4 (±1.3) | 90.2 (±1.3) | 58.6 (±1.0) | 57.6 (±1.3) | 53.6 (±0.7) | 56.1 (±1.0) |
| | All Joints FDE | 137.7 (±2.1) | 134.3 (±1.9) | 110.5 (±1.7) | 106.7 (±2.0) | 101.7 (±1.5) | 105.1 (±1.6) |
| | Wrists ADE | 107.3 (±1.5) | 114.8 (±1.3) | 82.5 (±1.2) | 79.6 (±1.4) | 76.7 (±1.0) | 74.8 (±1.0) |
| | Wrists FDE | 185.5 (±2.6) | 165.0 (±2.2) | 151.1 (±2.2) | 143.1 (±2.3) | 140.9 (±2.1) | 138.8 (±2.0) |

Table 2: Average forecast metrics (in mm) for all models across all datasets.

| Model → | | GROUNDTRUTH | | BASELINES | | LEARNING-BASED | | | | | |
|---|---|---|---|---|---|---|---|---|---|---|---|
| Metric ↓ | | CUR | FUT | CVM | WORST | BASE | SCRATCH | FINETUNED | MANICAST-T | MANICAST | MANICAST-W |
| **REACT** | Stop Time (ms) | 0 (±0) | 1000 (±0) | 367.8 (±50.9) | 1000 | -138.9 (±18.8) | -237.5 (±54.4) | 203.3 (±22.3) | 271.1 (±25.6) | 246.7 (±25.8) | 290.0 (±29.9) |
| | Restart Time (ms) | 0 (±0) | 1000 (±0) | 235.6 (±18.2) | 0 (±0) | 256.7 (±45.6) | 235.0 (±16.8) | 441.1 (±30.8) | 454.4 (±36.7) | 455.6 (±33.5) | 496.7 (±27.9) |
| | FDR | 0% | 0% | 67% | 100% | 11% | 11% | 0% | 11% | 0% | 11% |
| **HAND.** | Goal Detection (ms) | 0 (±0) | 1000 (±0) | 246.0 (±146.0) | - | -127.0 (±16.6) | -61.6 (±23.6) | 189.3 (±47.8) | 473.3 (±123.7) | 450.0 (±124.8) | 488.4 (±133.4) |
| | Correct Goal Rate | 100% | 100% | 20% | 0% | 90% | 80% | 100% | 100% | 100% | 100% |
| | Path Length (mm) | 459.0 (±37.0) | 381.1 (±39.8) | 485.0 (±85.0) | - | 488.9 (±56.2) | 466.2 (±33.2) | 432.0 (±39.0) | 428 (±42.0) | 404.0 (±45.0) | 436.0 (±49.4) |
| | Time to Goal (s) | 4.59 (±0.37) | 2.87 (±0.24) | 3.77 (±0.86) | - | 4.32 (±0.48) | 4.60 (±0.68) | 3.87 (±0.28) | 3.91 (±0.32) | 3.96 (±0.54) | 4.17 (±0.45) |

Table 3: We integrate forecasts of different models into STORM for the reactive stirring and object handover tasks. The standard error for each planning metric is shown inside parentheses.

WRISTONLY-T has lower wrist ADE than WRISTONLY during transition windows. As expected, All Joints ADE for these models are significantly higher than FINETUNED and MANICAST. While this might be fine for the reactive stirring and handover task, we cannot use this model for the collaborative table-setting task. Such an approach would generally be limited to tasks solely relying on wrist forecasting.

## A.4 Proof for Lemma 1

*Proof.* We first analyze the performance of a model trained on $P(\phi)$.

Let's assume that a model trained with MLE loss on $P(\phi)$ bounds the average L1 distance between the ground truth distribution $P(\xi_H|\phi)$ and the learned distribution $P_\theta(\xi_H|\phi)$ on $P(\phi)$ by $\varepsilon$, i.e.

$$\sum_\phi P(\phi) \sum_{\xi_H} |P(\xi_H|\phi) - P_\theta(\xi_H|\phi)| \leq \varepsilon$$

Then, for any given $\xi_R$ be the robot trajectory $\xi_R$, the final loss $\ell(\theta)$, i.e. the expected cost difference of $\xi_R$ due to the ground truth distribution and the forecast model can be expressed as:

$$\ell(\theta) = \sum_{\phi} P(\phi) \left( \left| \sum_{\xi_H} C(\xi_R|\xi_H) P(\xi_H|\phi) - \sum_{\xi_H} C(\xi_R|\xi_H) P_\theta(\xi_H|\phi) \right| \right)$$

$$= \sum_{\phi} P(\phi) \left( \left| \sum_{\xi_H} C(\xi_R|\xi_H) \left( P(\xi_H|\phi) - P_\theta(\xi_H|\phi) \right) \right| \right)$$

$$\leq \sum_{\phi} P(\phi) \left( \| C(\xi_R|\xi_H, \phi) \|_\infty \sum_{\xi_H} |P(\xi_H|\phi) - P_\theta(\xi_H|\phi)| \right) \qquad \text{(Holder's ineq.)}$$

$$\leq \sum_{\phi} P(\phi) \left( C_{\max}(\phi) \sum_{\xi_H} |P(\xi_H|\phi) - P_\theta(\xi_H|\phi)| \right)$$

$$\leq \max_{\phi} C_{\max}(\phi) \sum_{\phi} P(\phi) \left( \sum_{\xi_H} |P(\xi_H|\phi) - P_\theta(\xi_H|\phi)| \right)$$

$$\leq C_{\max} \varepsilon$$

where $C_{\max}(\phi) = \|C(\xi_R|\xi_H, \phi)\|_\infty$ is the maximum cost of a robot trajectory given a context, $C_{\max} = \max_\phi C_{\max}(\phi)$ is the maximum cost across all context. $C_{\max}$ can be high in general, resulting in an inflated bound for the model above.

Now let's assume we train a model to minimize loss on the new distribution $Q(\phi) = 0.5P(\phi) + 0.5P_T(\phi)$ and get the following bound

$$\sum_{\phi} Q(\phi) \sum_{\xi_H} |P(\xi_H|\phi) - P_\theta(\xi_H|\phi)| \leq \varepsilon$$

Then the loss can be expressed as:

$$\ell(\theta) = \sum_{\phi} P(\phi) \left( \left| \sum_{\xi_H} C(\xi_R|\xi_H) P(\xi_H|\phi) - \sum_{\xi_H} C(\xi_R|\xi_H) P_\theta(\xi_H|\phi) \right| \right)$$

$$\leq \sum_{\phi} P(\phi) \left( C_{\max}(\phi) \sum_{\xi_H} |P(\xi_H|\phi) - P_\theta(\xi_H|\phi)| \right)$$

$$\leq \sum_{\phi} Q(\phi) \frac{P(\phi) C_{\max}(\phi)}{Q(\phi)} \sum_{\xi_H} |P(\xi_H|\phi) - P_\theta(\xi_H|\phi)|$$

$$\leq \max_{\phi} \frac{P(\phi) C_{\max}(\phi)}{Q(\phi)} \sum_{\phi} Q(\phi) \sum_{\xi_H} |P(\xi_H|\phi) - P_\theta(\xi_H|\phi)|$$

For $Q(\phi) = 0.5P(\phi) + 0.5P_T(\phi)$, we need to bound the ratio

$$\max_{\phi} \frac{P(\phi) C_{\max}(\phi)}{Q(\phi)} = \max_{\phi} \frac{P(\phi) C_{\max}(\phi)}{0.5P(\phi) + 0.5P_T(\phi)}$$

There are two cases to consider:

**Case 1**: $C_{\max}(\phi) \leq \delta$. Then $P_T(\phi) = 0$, and the ratio is bounded by

$$\max_{\phi} \frac{P(\phi) C_{\max}(\phi)}{0.5P(\phi) + 0.5P_T(\phi)} \leq \frac{P(\phi)\delta}{0.5P(\phi)} \leq 2\delta$$

**Case 2**: $C_{\max}(\phi) \geq \delta$. Then ratio is maximized when $C_{\max}(\phi)$ is maximized for $\phi = \phi^*$

$$\max_{\phi} \frac{P(\phi) C_{\max}(\phi)}{0.5P(\phi) + 0.5P_T(\phi)} \leq \frac{P(\phi^*) C_{\max}(\phi^*)}{0.5P_T(\phi^*)} \leq \frac{P(\phi^*) C_{\max} \sum_{\phi} P(\phi) \mathbb{I}(C_{\max}(\phi) \geq \delta)}{0.5P(\phi^*)}$$

$$\leq 2C_{\max} \mathbb{E}_{P(\phi)} [\mathbb{I}(C_{\max}(\phi) \geq \delta)]$$

Combining these cases, we can bound the ratio $\max_\phi \frac{P(\phi)C_{\max}(\phi)}{Q(\phi)}$ as

$$\max_\phi \frac{P(\phi)C_{\max}(\phi)}{Q(\phi)} \leq 2\max(\delta, C_{\max}\mathbb{E}_{P(\phi)}[\mathbb{I}(C_{\max}(\phi) \geq \delta)])$$

The ratio above can be no worse than $C_{\max}$ by a factor of 2, and can be much smaller based on the choice of $\delta$. Intuitively setting $\delta$ to be very high makes the transition probability $P_T(\phi)$ peaky driving down the second term, while making $\delta$ to be small makes the transition probability close to the original distribution, driving down the first term.

$\square$

## A.5  MPC Planner Details

We use the open-sourced STORM codebase[3] to implement sampling-based model-predictive control on a 7-DOF Franka Research 3 robot arm. At every timestep, the planner samples robot trajectories and evaluates the cost function with MANICAST forecasts. The robot executes the first action from the lowest-cost plan and updates its sampling distribution for the next timestep using the MPPI [71] algorithm. The manipulation components of the cost function independent of the human remain unchanged. We additionally introduce a collaborative task-specific cost component ($T(\xi_R|\hat{\xi}_H)$) that depends on the future human trajectory. The cost function optimized by the planner is laid out in Eq.3. Self-collisions are checked by training the jointNERF model introduced by Bhardwaj et al. [70].

$$C(\xi_R|\hat{\xi}_H) = \alpha_s\hat{C}_{stop}(\xi_R) + \alpha_j\hat{C}_{joint}(\xi_R) + \alpha_m\hat{C}_{manip}(\xi_R) + \alpha_c\hat{C}_{coll}(\xi_R) + \boldsymbol{\alpha_t}\mathbf{T}(\boldsymbol{\xi_R}|\hat{\boldsymbol{\xi}}_\mathbf{H}) \tag{3}$$

## A.6  Tasks for Collaborative Manipulation

We describe three collaborative manipulation tasks that focus on house-hold cooking activities.

**Reactive Stirring**: In this cooking task, the human and robot share a common workspace. While the robot arm is performing a stirring motion, the human may add vegetables into the pot. The robot arm preemptively predicts the arrival of the human arm and retracts back to give the human arm sufficient space to reach into the pot. The task-specific component of the cost function is:

$$T(\xi_R|\hat{\xi}_H) = \sum_{t=1}^{T} \mathbb{1}\left[D(\hat{s}_t^H, s^{pot}) \leq \varepsilon\right]\|s_t^R - s^{rest}\| + \mathbb{1}\left[D(\hat{s}_t^H, s^{pot}) > \varepsilon\right]\|s_t^R - \xi_{stir}^t\| \tag{4}$$

The cost function checks whether the human's position ($\hat{s}_t^H$) is close to the pot's position ($s^{pot}$) and decides whether to move to a pre-defined resting position ($s^{rest}$) or to continue stirring in a circular trajectory ($\xi_{stir}$) starting from the current state of the robot ($s_0^R$). A cost-aware forecasting model for this task should be able to predict the arrival and departure of the human ahead of time.

**Human-Robot Handovers**: Handovers of objects are an important task in the kitchen. When a human is handing over an object, a robot arm should move towards the intended handover location. The task-specific component can be described as:

$$T(\xi_R|\hat{\xi}_H) = \sum_{t=1}^{T} \mathbb{1}\left[IsObjectInHand(s_0^H)\right]\hat{C}_{pose}\left(X_t^{ee}, GraspPose(X_0^{ee}, \hat{X}_T^{H_{wrist}})\right) \tag{5}$$

Similar to prior work [56], the robot motion is initiated when the human arm has picked up the handover object. The robot's end-effector ($X_t^{ee}$) moves towards a grasp location that is computed using the final wrist position of the human ($\hat{X}_T^{H_{wrist}}$). The orientation of the grasp pose is calculated by drawing a straight line from the current end-effector position ($X_t^{ee}$) to the grasp location.

**Collaborative Table Setting**: Movements on top of a table in the presence of a human in the workspace are a common collaborative manipulation task. Motion planners should not only avoid

---

[3]https://github.com/NVlabs/storm

collision in the current timestep but also be able to forecast future motion and preemptively avoid collisions with the human body. The cost function is simply given by:

$$T(\xi_R|\hat{\xi}_H) = \sum_{t=1}^{T} \hat{C}_{pose}\left(X_t^{ee}, X_t^G\right) + \beta \hat{C}_{coll}\left(s_t^R, s_t^H\right) \tag{6}$$

Here, $\beta$ is the relative weight given to the collision avoidance component compared to the goal-reaching component. Collisions are checked between the human body and robot arm by representing them as a pack of sphere and cuboid rigid bodies.

### A.7 Collaborative Manipulation Dataset (CoMaD)

Similar to a real-world collaborative activity, in much of the episode, both humans perform their respective cooking tasks in isolation. Episodes of reactive stirring and handovers contain 3-5 close-proximity interactions, each of which are short (4-5 seconds) compared to the length of the overall episode (30-60 seconds). Often, these interactions are initiated by verbal requests or subtle facial gestures. Collaborative table setting consists almost entirely of close-proximity fast human arm movements. We collect an RGB visual view of the scene containing audio along with motion capture data of both humans' upper bodies. We also annotate transition windows for interactions in each episode.

### A.8 Model Implementational Details

We train our forecasting models using the STS-GCN [46] architecture on an upper body skeleton consisting of 7 joints (Wrists, Elbows, Shoulders, and Upper Back). The last 0.4 seconds (10 timesteps) of motion is input to the models and the next 1 second (25 timesteps) of motion is predicted. We pretrain for 50 epochs on AMASS (1 hour) and finetune on CoMaD for 50 epochs (5 minutes). We divided the episodes in CoMaD into train, validation, and test sets (8:1:1).

