# OpenReview forum: "ManiCast: Collaborative Manipulation with Cost-Aware Human Forecasting"
_robot-learning.org/CoRL/2023/Conference — CoRL 2023 Poster_

### Official Review · Reviewer_EFoV · 2023-07-18

**Confidence:** 3
**Originality:** Good
**Technical Quality:** Fair
**Clarity Of Presentation:** Very Good
**Impact:** 3

**Recommendation:**

Weak Accept: I recommend accepting the paper, but will not argue for my recommendation if the majority of other reviewers have a different opinion.

**Review:**

Strengths:

1. Very interesting idea and the task-oriented prediction loss is technically sound and novel.

2. The results support the hypothesis that cost-based loss can improve prediction for planning.

3. Writing is clear.

Weaknesses:

1. My main concern is that it is unclear if there could be a simpler approach that can achieve the same result. Looking at the pose prediction errors, it is clear that the *finetuned* model using MLE loss produces the lowest error on all joints, but the proposed model has better accuracy for wrists, which I assume are more important for planning. However, would adding more weights to the MLE loss on wrists give you a similar benefit? Or could you simply only train to predict wrists and fit a typical pose to the predicted wrists' positions?

2. On a related note, there should be more analyses/discussions on the performance comparison between the *finetuned* baseline and ManiCast. E.g., I do not think it is accurate to say ManiCast has a more accurate prediction than *finetuned*. Its overall error on all joints is higher than *finetuned*.


**Quality Of The Limitations Section:**

Limitations are addressed clearly

**Questions For Rebuttal:**

Please address my concerns above.

**Robotics Focus:**

Sufficient demonstration on hardware

**Summary Of Paper:**

This paper presents an alternative learning objective for human pose forecasting for human-robot manipulation. Instead of only using pose prediction loss, it uses cost prediction loss to finetune the prediction model. The model was evaluated on three tasks -- Reactive Stirring Object Handover Collaborative Table Setting. Experimental results show that the prediction results, though not more accurate in terms of errors for all joints, are more accurate for planning.

**Summary Of Recommendation:**

My main concern is that if a simpler approach is essentially enough to achieve a similar result. If this is adequately addressed, I am willing to increase my rating.

---

### Official Review · Reviewer_Z1jy · 2023-07-19

**Confidence:** 4
**Originality:** Fair
**Technical Quality:** Fair
**Clarity Of Presentation:** Fair
**Impact:** 2

**Recommendation:**

Weak Reject: I recommend rejecting the paper, but will not argue for my recommendation if the majority of other reviewers have a different opinion.

**Review:**

Quality:

The quality of the work is generally fine. The paper addresses an important problem in robotics, namely seamless human-robot collaboration in close proximity tasks. The authors present a well-defined framework, ManiCast, that combines cost-aware human forecasts with a model predictive control planner. The experiments are conducted on real-world tasks, and the proposed framework is compared against various baselines, providing an evaluation of its performance.

Clarity:

The paper is generally well-written. The introduction sets the context and motivation for the research, and the problem formulation is clearly presented. The authors provide a fair explanation of the proposed framework, including the training and inference processes. However, the wording of some of the premises might be restructured to align the contributions with the achieved outcomes. Also, some sections, particularly in the related work, could benefit from clearer organization and more explicit connections to the authors' work. Additionally, the presentation of the evaluation results could be further improved by providing more specific details and analysis.

Originality:

The paper introduces a novel approach to collaborative manipulation by focusing on cost-aware human forecasts. While there is existing research on human-robot collaboration and human motion forecasting, the combination of these two aspects in the proposed framework is original. The authors integrate state-of-the-art pre-trained human forecast models with an existing model predictive control planner, showcasing the effectiveness of their approach. The upsampling of transition points and the dimension weighting strategies contribute to the originality of the work.

Significance:

The work presented in the paper has some implications for the field of robotics, specifically in the area of collaborative manipulation. The ability to accurately forecast human motion while considering the cost of robot plans is crucial for safe and efficient human-robot interactions. The proposed framework, ManiCast, addresses this challenge but falls short on providing convincing evaluations in some aspects. The introduction of the CoMaD dataset is valuable but its coverage does not seem to be large enough to be effectively used by the community.

Strengths:

- The integration of cost-aware human forecasts with a model predictive control planner is a novel and valuable contribution to the field.

- The use of real-world tasks and the creation of the CoMaD dataset enhance the practical relevance of the research.

Weaknesses:

- The contribution claims are not well-supported by what is presented in the paper. I agree with the first contribution, i.e., the introduction of the combined forecasting and MPC-based planner. However, the introduced dataset only includes two persons (unclear backgrounds, biases, etc.) and three tasks which makes it hardly a full-scale dataset that can be used by the community for diverse purposes. Similarly, the other contribution on ‘extensive real-world evaluation’ also falls short considering the coverage of tasks and evaluations reported.

- The paper emphasizes, as one aspect of its novelty, the forecasting of the full upper body. However, the tasks considered in the experiments are not convincing enough to understand the need for such upper body movement prediction.

- Close-proximity interactions are emphasized, however, the two tasks (handover, stirring), due to setups being investigated, do not seem to include such a close interaction that is expected to impose harder motion planning constraints on the robot partner. The third task, table setting, has not been investigated thoroughly.

- Some sections, particularly in the related work, could benefit from clearer organization and better contextualization within the authors' work.
  - There are some works in the literature that combines human hand/arm motion prediction together with robot motion planning and/or action selection (some examples {1,2}, but there are definitely more) which have not been covered.
  - My concern is less on the coverage of additional papers but more on how some premises are phrased: e.g., “To the best of our knowledge, this is the first paper to combine state-of-the-art pre-trained human forecast models like STS-GCN [3] with a real-time MPC planner [4] for collaborative human-robot manipulation tasks.” Yes this work might be the first to combine STS-GCN and an MPC planner, but this sentence sounds also a little bit like this is the first work to combine human motion prediction and a robot planner. I’d suggest rephrasing these sentences to emphasize that the new parts are the used methods, not the formulation of a predictor and a reactive planner. Similar arguments provided in the Related Work should be reconsidered, and if necessary, rephrased.
  - “...to the best of our knowledge, our work is the first to integrate an entire upper body forecast into a robot manipulation planner.” But what is necessary/good about upper body movement prediction?
  - “In contrast to these methods which focus on specific motions such as reaching and consider certain parts of the body, our work uses a history of human poses to predict future poses of the entire upper body in real-time for safe planning around the human body.” This movements in this work are also reaching motions, and other works also consider the history, so these premises are not well-supported in my opinion.


- The experiments are not extensive.

- The presentation of the evaluation results could be more detailed and include deeper analysis and interpretation.

  - The BASE and SCRATCH baselines are meaningful, but to better evaluate the proposed approach a simple learning-based baseline may be implemented (e.g., just a plain recurrent neural network policy)

  -  Table 1 could be placed in the appendix. The main text would be better off with a summarized version of this data.

  - Similarly, Table 2 could include only the important bits. And what is presented in this table should be explained clearly. As far as I understood what is being considered is the difference between the restart time and stop time for the stirring task but it’s not clearly explained. Why should we not care about the response time for stopping, but just the difference? And also is it well justified to have a higher FDR when the reaction time is faster for MANICAST-W?

  - Overall, a good chunk of data in those tables have not been discussed, nor explained.

  - Figure 3: Is the unit of time in the x-axis in seconds? A reaching motion takes ~10seconds?

  - why isn’t any detailed analysis of the table setting task? It seems to be a more representative close-proximity task where motion prediction and reactive robot motion planning would be more critical.

  - The upper body emphasis is not well analyzed and supported by the evaluations. Wrist seems to have the most influence, that is to be expected due to the nature of the task setups which do not have much close-proximity collaborative motions.
    - How about just using the wrist prediction?

  - It’s not clear why joint positions are predicted.
    - How do you enforce kinematic consistency?
    - Have you considered joint angle predictions given a kinematic model, which you can get from the motion tracker? Then the output dimensionality would be much smaller as well.


_1. “Goal Set Inverse Optimal Control and Iterative Re-planning for Predicting Human Reaching Motions in Shared Workspaces”, Jim Mainprice, Rafi Hayne, Dmitry Berenson, 2016, TRO._

_2. “An Ontology for Human-Human Interactions and Learning Interaction Behavior Policies”, O S Oguz, W Rampeltshammer, S Paillan, D Wollherr, 2019, T-HRI._

**Quality Of The Limitations Section:**

Additional details required

**Questions For Rebuttal:**

As highlighted in the review section:

- Clarify and rephrase the contribution claims to better align with the presented work and its outcomes.

- Provide a more thorough explanation and justification for the emphasis on upper body movement prediction.

- Consider selecting evaluation tasks that involve more representative close-proximity interactions to better demonstrate the effectiveness of the proposed framework.

- Improve the organization and contextualization of the related work section, ensuring clear connections to the authors' work and addressing any missing relevant references.

- Conduct a more extensive evaluation with a variety of tasks and provide a deeper analysis and interpretation of the results.

- Consider including a simple learning-based baseline for comparison to better evaluate the proposed approach.

- Streamline and clarify the presentation of evaluation results, including Tables 1 and 2, and provide detailed explanations for the data and analysis.

- Analyze and discuss the table setting task, as it represents a more critical scenario for motion prediction and reactive robot motion planning.

- Revisit the justification for joint position prediction and consider exploring joint angle predictions given a kinematic model, which could reduce output dimensionality.

- Consider incorporating more explicit analysis of the impact and significance of wrist predictions, which appear to be the most influential in the evaluation tasks.


**Robotics Focus:**

Sufficient demonstration on hardware

**Summary Of Paper:**

The paper introduces ManiCast, a framework for collaborative manipulation in robotics that focuses on generating cost-aware human forecasts. The main idea is to produce forecasts that capture how future human motion would affect the cost of a robot's plan, rather than predicting the most likely human motion. It combines pre-trained human forecast models with a real-time model predictive control planner.
As the main contribution, the paper presents a novel approach to collaborative manipulation in robotics, emphasizing cost-aware human forecasts as a means to enable seamless interactions and improve planning performance in close proximity tasks.
The framework is evaluated through real-world tasks (reactive stirring, object handovers, and collaborative table setting), and compared against a range of learned and heuristic baselines. A new dataset comprising human-human collaborative manipulation on three kitchen tasks is also introduced.

**Summary Of Recommendation:**

While the work addresses an important problem in robotics, and proposes a novel adaptation of the loss function for motion prediction, the review identifies several weaknesses in the paper, including the lack of convincing evidence to support the claims, the need for major revisions to improve clarity and organization, and uncertainty regarding the impact of the work. The reviewer suggests that significant additional evidence, thorough revisions, and more compelling evaluations are necessary to strengthen the paper.

---

### Official Review · Reviewer_M4Pj · 2023-07-20

**Confidence:** 3
**Originality:** Very Good
**Technical Quality:** Very Good
**Clarity Of Presentation:** Very Good
**Impact:** 3

**Recommendation:**

Weak Accept: I recommend accepting the paper, but will not argue for my recommendation if the majority of other reviewers have a different opinion.

**Review:**

I think the paper is well written and is clear. It is a combination of human prediction works and human robot safety works. I think that there is valuable insights on how to train the ManiCast with importance sampling and dimension weighting to make this work because the human's wrist's direction is important for handling handover tasks. I think it would have been nice to have a video, but I guess it is difficult to blur everyone out for anonymity. That is the only weakness so far as I would like to see the system work in real time on video.

**Quality Of The Limitations Section:**

Limitations are addressed clearly

**Questions For Rebuttal:**

1. Needs a supporting video to show the real time operation of the human prediction and planner all working together on the Franka Research 3.

**Robotics Focus:**

Sufficient demonstration on hardware

**Summary Of Paper:**

The authors in this paper first collected a dataset called CoMaD which contains motion capture data of two people doing 3 collaborative cooking tasks. Then the authors introduce ManiCast which is an algorithm for forecasting future human motion and then adjusting the planning cost for the robot in regions that are close to the human in case it leads to a collision. ManiCast is able to run in realtime at 120 Hz to update the robot's MPC planner at 50 Hz.

**Summary Of Recommendation:**

I decided on a weak accept because there is no video. I need more proof that the method is able to plan faster around humans while ensuring safety compared to other previous human robot safety papers from researchers like Professor Changliu Liu which you have cited none of.

---

> ### Comment · Reviewer_M4Pj · 2023-08-14
> **Response to Rebuttal**
>
> Sorry we responded to the area chair's post, but I don't think the authors can see it so I copied my response below:
>
> I have watched the videos and appreciate the evidence of the real-time online execution. I am still in favor of the paper as a weak accept. While the results may not be outstanding, I think the authors have done enough experiments to illustrate the efficacy of their approach in the real world. I, however, am not an expert in the robot safety and planning prediction space, so I do not know if these results are state of the art currently.

---

### Official Review · Reviewer_1c4Z · 2023-07-20

**Confidence:** 4
**Originality:** Good
**Technical Quality:** Good
**Clarity Of Presentation:** Good
**Impact:** 3

**Recommendation:**

Weak Accept: I recommend accepting the paper, but will not argue for my recommendation if the majority of other reviewers have a different opinion.

**Review:**

The paper is well-structured, effectively introducing the problem domain and motivating the need for cost-aware human forecasting in collaborative manipulation tasks. The experimental results presented clearly demonstrate the advantages of the cost-based strategy employed in the proposed framework. The use of task-oriented forecasting loss is justified and delivers a clear message to readers.

However, there are certain weaknesses that need to be addressed. In Table I, the "Average forecast metrics" show that the Base model produces the lowest error on all joints, while the proposed model exhibits better accuracy for wrists. It would be beneficial to elaborate on whether the proposed model assigns more weight to the wrists during training or solely focuses on predicting wrists while fitting a typical pose to their positions. Providing further analysis and discussion on this aspect would strengthen the paper.

Additionally, more detailed analyses and discussions are required to compare the performance of the Finetuned baseline and ManiCast. The overall error of ManiCast on all joints appears higher than that of the Finetuned baseline. Expanding on this comparison would provide valuable insights and help readers understand the relative performance of the two approaches.

Another important consideration is the comparison between the proposed approach and a simple planning scheme for the stirring task. It is essential to evaluate whether the proposed approach can outperform a basic planning scheme that involves detecting the stirring place, moving to that location, and stirring while avoiding human wrists. Since all the actions in this task are deterministic, it is crucial to assess whether the proposed approach offers advantages over such a basic planning scheme.

**Quality Of The Limitations Section:**

Additional details required

**Questions For Rebuttal:**

Can the authors explain whether the improved accuracy for wrists is a result of assigning more weight to wrists during training or due to a specific focus on predicting wrists with a typical pose?
The review mentions that the overall error of ManiCast on all joints appears higher than that of the Finetuned baseline. Can the authors provide a detailed analysis and discuss the reasons behind this difference in performance?
How does the proposed approach compare to a basic planning scheme for the stirring task, which involves detecting the stirring place, moving, and stirring while avoiding human wrists? Can the authors evaluate whether the proposed approach outperforms this simple scheme and provide insights into the relative strengths and limitations?

**Robotics Focus:**

Sufficient demonstration on hardware

**Summary Of Paper:**

The paper "ManiCast: Collaborative Manipulation with Cost-Aware Human Forecasting" proposes a framework that integrates cost-aware human forecasts into a model predictive control planner for collaborative manipulation tasks. This review aims to provide constructive feedback to improve the scientific writing and address specific aspects of the paper.

**Summary Of Recommendation:**

In conclusion, the paper "ManiCast: Collaborative Manipulation with Cost-Aware Human Forecasting" presents a valuable framework for incorporating cost-aware human forecasts into collaborative manipulation tasks. The strengths of the paper lie in its well-written structure, the clear advantages of the cost-based strategy, and the justification for task-oriented forecasting loss. Addressing the weaknesses by providing more in-depth analysis and comparisons, as suggested, would significantly enhance the overall scientific contribution of the paper.

---

### Author Response · Authors · 2023-08-08
**Global Response (addressing common concerns) 1/3**

We thank all the reviewers for the helpful comments. We are pleased to hear that the reviewers find our work well-written (R1, R2, R3, R4). Our work introduces ManiCast, a cost-aware human forecasting model for collaborative robot manipulation. ManiCast does this by upsampling transition periods and upweighting wrist errors in its loss function. We are encouraged that reviewers acknowledge the advantages of the cost-aware forecasting loss formulation (R1), and agree our contribution is novel (R3, R4) and a valuable contribution (R3) to the community. Further, our experimental results support the effectiveness of our framework for collaborative manipulation planning (R1, R4) and draw valuable insights (R2).

We considered three collaborative manipulation tasks: reactive stirring, object handovers, and collaborative table setting. In response to R2’s request, we demonstrate the effectiveness of our approach by attaching three videos in each rebuttal response of real-time execution of our Franka Research 3 with an anonymized human partner on each task. Our system can track a human user's pose, forecast their movements, and control a robotic arm in real-time and at high speed. Our system's forecasting module runs at 120Hz, used by an MPC at 50Hz.

**Update 08/09: We include an [online link](https://drive.google.com/drive/folders/191Uobt4szUk3YErOmBR9y2S7jGfTbd4g?usp=sharing) to view the videos.**

Further, we resolved the common concerns by uploading a revised paper draft and appendix. We introduce the models presented in the following discussions:

1) **FineTuned**: Trained on MLE loss.
2) **FineTuned-W**: Trained on MLE loss upweighting the wrist dimensions.
3) **ManiCast**: Trained by upsampling the transition data.
4) **Manicast-W**: Trained by upsampling the transition data and upweighting the wrist dimensions.
5) **WristOnly**: Trained only to predict wrist dimension with MLE loss; other joints use the last observable position.
6) **WristOnly-T**: Trained only to predict wrist dimension by upsampling the transition data; other joints use the last observable position.

---

> ### Author Response · Authors · 2023-08-08
> **Global Response (addressing common concerns) 2/3**
>
> **Q1: Analyzing the effects of wrist prediction and their importance in the planning tasks.**
>
> A: We found that the ManiCast (upsample transitions) and ManiCast-W (upsample transitions + upweights wrist) methods were roughly equally effective for predicting wrist error during transition periods (41.0mm vs. 40.7mm T-Wrists-ADE in the Handover Task and 67.1mm vs. 62.9mm T-Wrists-ADE in the Reactive Stirring Task). However, we observed that ManiCast typically produces less jerky forecasts (quantified by the 0% vs. 11% false detection rate in the reactive stirring task - Table 2). ManiCast models perform well on the wrists due to upweighting transition periods between motions (e.g., chopping a vegetable and putting it in the pot). These transitions correspond to large amounts of wrist movement. These transitions also yield the highest costs to the robot's motion, and learning to predict the movement of relevant joints in those moments is especially important and handled better than any baselines with our method.
>
> We also update the appendix with experiments (A.1 and A.2) to measure the effects of upweighting wrist error with different weights. We also experiment with a version of the FineTuned-W model that upweights wrist errors (MLE + Wrist Weight) without upsampling transitions. Observing Fig 1 in Appendix A.1, All ManiCast-W (upsample transitions  + upweight wrist) have lower all joints and wrists ADE in the transition periods, which are most relevant for planning performance. Hence, instead of simply focusing on the wrist joints, upsampling transition data is critical for the success of our framework.
>
> Further, we note a trade-off exists between all joints’ and wrists’ forecasting errors. As expected, increasing the wrist weight in ManiCast-W reduces prediction errors on the wrist joints but at the same time, the prediction errors on all joints increase. Furthermore, the decrease in wrist errors eventually plateaus around a wrist weight of 5, which justifies this hyperparameter choice for the ManiCast-W model presented in the main paper.
>
> Finally, we consider the simpler problem of only predicting the wrist joint position in forecasting (A.2). We introduce a new model called WristOnly and WristOnly-T that takes as input the entire upper body history and outputs wrist prediction (equivalent to setting wrist weights to be infinity in FineTuned-W and ManiCast-W). We construct the rest of the upper body by assuming the upper body at the last observable timestep remains still in the future. We observe that wrist forecasting errors for the WristOnly and WristOnly-T models are similar to Finetuned-W and ManiCast-W that also predict other upper body joints. We will still find that upsampling transition data points helps reduce the forecasting errors. In Fig 2 of the Appendix, we observe that WristOnly-T has lower wrist ADE than WristOnly (85.8mm vs 92.4mm) during transition windows. As can be expected, All Joints ADE of WristOnly is significantly higher than ManiCast. While this is acceptable for the reactive stirring and handover task, this model does not apply to the collaborative table-setting task. Such an approach would generally be limited to tasks solely relying on wrist forecasting, whereas ManiCast is generalizable to more complex tasks.

---

> > ### Author Response · Authors · 2023-08-08
> > **Global Response (addressing common concerns) 3/3**
> >
> > **Q2: Why do ManiCast models have higher errors on all joints than the FineTuned model?**
> >
> > A: Although, on average, the ADE and FDE values are higher for ManiCast compared to fine tuned model, Manicast performs significantly better in "transition" periods where accurate forecasts matter. Notably these are high-cost time slices where the human and robot enter close proximity due to a transition in the human's activities. We have updated Table 1 to also include forecasting errors for all joints during transition periods and refer to this updated table for our following discussion.
> >
> > In the Reactive Stirring Task, ManiCast has slightly higher All Joints ADE (45.1mm vs. 44.8mm) and Wrists ADE (67.1mm vs. 64mm). However, in the transition periods, ManiCast has lower All Joints ADE (58.4mm vs 63.3mm) as well as lower Wrists ADE (89.5mm vs 94.9mm). We observe similar trends in the object handover task. The collaborative table setting task comprises almost entirely of transitions and we can see that ManiCast has lower ADE than the Finetuned for wrists as well as all joints.
> >
> > In Figure 3, we show a typical reactive stirring episode. The total duration of the transition periods is significantly smaller than the entire duration of the episode. While a FineTuned model that maximizes the MLE loss has lower errors on average, it fails to forecast accurate human motion during critical transition periods. This leads to worse planning performance, as shown in Table 2. In the reactive stirring task, ManiCast detects the arrival of the human pose into the workspace earlier than the FineTuned model (+246.7ms vs. +203.3ms). In the object handover time, ManiCast can predict the handover location earlier than the FineTuned model (+450.0ms vs. +189.3 ms - Goal Detection Time).
> >
> > ~We are still incorporating feedback in our introduction and related work sections. We will share another updated draft imminently.~
> >
> > **Update 08/09: We updated our draft to incorporate feedback in our introduction and related work sections**

---

### Decision · Program_Chairs · 2023-08-30

**Decision:**

Accept (Poster)

**Comment:**

The paper introduces a novel framework for robotics that emphasizes the integration of cost-aware human forecasts into a model predictive control (MPC) planner.

Instead of predicting the most likely human motion, ManiCast aims to understand how future human movements might impact the robot's planning costs. This approach enhances collaborative tasks by enabling smoother interactions, especially in close proximity. The authors evaluated the framework's efficiency through real-world tasks and introduces a new dataset on human-human collaborative manipulation in kitchen scenarios. This review seeks to provide feedback to enhance the paper's scientific presentation and address its specific elements.

The majority of the reviewers agreed on the merits and technical soundness of the paper, and they found the author response helpful and convincing. The reviewers unanimously recommend accepting the paper (though `Z1jy` forgot to change the rating in the system). I recommend accepting this paper. The authors are encouraged to address the remaining issues, such as "1) effectiveness of predicting the full upper-body marginal,  and 2) tasks being considered are not really close-proximity tasks".